# Cuffless Blood Pressure Estimation with Confidence Intervals using Hybrid Feature Selection and Decision Based on Gaussian Process

Soojeong Lee [1,*], Gyanendra Prasad Joshi [1], Anish Prasad Shrestha [1], Chang-Hwan Son [2,*], Gangseong Lee [3,*]

[1] Department of Computer Engineering, Sejong University, 209 Neungdong-ro, Gwangjin-gu, Seoul 05006, Republic of Korea; joshi@sejong.ac.kr (G.P.J.); anishpshrestha@sejong.ac.kr (A.P.S.)

[2] Department of Software Science & Engineering, Kunsan National University, 558 Daehak-ro, Gunsan-si 54150, Republic of Korea

[3] Ingenium College, Kwangwoon University, 20 Kwangwoon-ro, Nowon-gu, Seoul 01897, Republic of Korea

[*] Correspondence: leesoo86@sejong.ac.kr (S.L.); cson@kunsan.ac.kr (C.-H.S.); gslee0115@gmail.com (G.L.)

**Abstract:** Cuffless blood pressure (BP) monitoring is crucial for patients with cardiovascular disease and hypertension. However, conventional BP monitors provide only single-point estimates without confidence intervals. Therefore, the statistical variability in the estimates is indistinguishable from the intrinsic variability caused by physiological processes. This study introduced a novel method for improving the reliability of BP and confidence intervals (CIs) estimations using a hybrid feature selection and decision method based on a Gaussian process. F-test and robust neighbor component analysis were applied as feature selection methods for obtaining a set of highly weighted features to estimate accurate BP and CIs. Akaike's information criterion algorithm was used to select the best feature subset. The performance of the proposed algorithm was confirmed through experiments. Comparisons with conventional algorithms indicated that the proposed algorithm provided the most accurate BP and CIs estimates. To the best of the authors' knowledge, the proposed method is currently the only one that provides highly reliable BP and CIs estimates. Therefore, the proposed algorithm may be robust for concurrently estimating BP and CIs.

**Keywords:** cuffless blood pressure estimation; confidence interval; Gaussian processing; hybrid feature selection and decision; F-test; Akaike's information criterion; robust neighbor component analysis; photoplethysmography

## 1. Introduction

By 2019, 523 million people have had a cardiovascular disease, and 18.6 million related deaths have been reported [1]. High blood pressure is a direct cause of death from cardiovascular disease (CVD) [2,3]. Therefore, accurate blood pressure (BP) measurements are necessary for diagnosing hypertension. BP monitoring has become vital for people with CVD, particularly elderly people who are living alone. Rapid changes in BP in these patients can indicate an underlying severe illness. Furthermore, BP varies owing to intrinsic physiological changes for various reasons, such as food intake, environmental temperature, exercise, disease, and stress. Thus, the precision and uncertainty of BP measurements induced by physiological parameters [4] have been a constant concern for clinicians and practitioners [5–7]. However, using BP monitoring devices for estimating BP uncertainty is currently impossible. This necessitates a standard protocol for confidence intervals (CIs), representing the uncertainty of BP monitors [5,8,9]. Most BP monitors offer only single-point estimates without CIs [5,8,9]. Therefore, patients, nurses, and physicians may be unable to differentiate intrinsic variations owing to the estimate's statistical variation and physiological processes [7–9].

Machine learning (ML) algorithms are commonly used for BP estimation [5,9–12]. ML algorithms, including multiple linear regression (MLR) [13,14], artificial neural networks (ANNs) [14–16], and support vector machine (SVM) [17–19] have been utilized for estimating BP [14,20–22]. Wang et al. [15] introduced a BP estimation method using photoplethysmography (PPG) signals using novel ANNs. Nandi et al. [23] proposed a new long short-term memory (LSTM) and convolutional neural network using cuffless BP estimation based on PPG and electrocardiogram (ECG) signals. Multichannel PPG was introduced using an SVM-ensemble-based continuous BP estimation in [19]. Qiu et al. [14] proposed a new method for estimating BP using a window-function-based piecewise neural network. This study evaluated a random-forest-based regression network, three-layer ANN-based regression network, and SVM model using PPG signals. Many studies on PPG-signal-based cuffless BP estimation have been conducted [13,15,21–23]. By contrast, studies on CIs estimation are limited. However, there have been studies on the uncertainty of estimating a few BP data based on conventional oscilloscope BP measurements [5–9]. The two most used methods for cuffless BP estimation were obtained using the extracted features and pulse transit time (PTT) from PPG signal pulses [14,24–26]. The feature extraction method using PTT effectively estimates BP because PTT is closely correlated with BP [27]. According to this principle, arterial pressure can be determined by measuring pulse wave velocity (PWV) at pulse wave speed. This is because changes in PTT correspond to changes in PWV at a fixed distance and indicates a change in BP [26,28–30].

Accordingly, this study proposed a new methodology for simultaneously estimating BP and CIs using cuffless BP measurement through a hybrid feature selection and decision based on the Gaussian process (GP). This study aimed to estimate BP and CIs, representing the uncertainty of cuffless BP estimation. Moreover, the GP can directly generate uncertainty estimates [31–33], such as providing a distribution of estimates rather than a single value. Another advantage of GP algorithms is that, similar to other kernel methods, they can be optimized precisely for given hyperparameter values [34]. Therefore, they perform well owing to the well-optimized parameter values, particularly with limited datasets [34]. However, the CIs automatically calculated by the GP algorithm are too wide to effectively represent the uncertainty. Therefore, CIs are estimated by applying the bootstrap method [8] using the results of the GP algorithm [31] to represent the uncertainty in the cuffless BP estimation.

The proposed hybrid feature selection and decision based on the GP algorithm can provide a means for distinguishing between estimation errors (statistical variance of estimates) and changes in estimates owing to physiological variability [5,7]. This study obtained CIs using a bootstrap algorithm [10,35] to determine the uncertainty (physiological variability) in cuffless BP estimation. Specifying CIs for cuffless measurements is beneficial because CIs measurement are necessary for estimating BP. If the BP measurement's CIs are too broad, healthcare workers may misjudge a patient's health status. Therefore, establishing CIs based on accurate BP estimates allows for more accurate and faster meaningful determinations of BP measurement CIs [9]. However, studies on determining the uncertainty of physiological measurements [4] using oscillometric BP signals [5–8] are limited. Therefore, a new method for evaluating and representing the uncertainty in BP measurement should be developed by providing an estimated range for cuffless BP measurements. Consequently, repeatable, irregular, and broad CIs based on aggregated statistical data can provide patients, clinicians, and families a warning system for BPs outside the normal range [5,9,10]. Another problem in improving the performance of ML algorithms is the choice of features to replace the original ones and using them as input data. Feature selection is an essential part of the learning algorithm's performance; it selects a subset of features with higher weights for the response variable and eliminates duplicate features [36,37]. Thus, the proposed methodology uses a hybrid F-test [38] and robust neighbor component analysis (RNCA) [39] to select weighted features from the original features. The best feature set is selected using Akaike's information criterion (AIC) [40] based on the maximum likelihood from the GP algorithm [31]. First, the F-test is used to acquire weighted features to compute

a feature's within- and between-group variance ratios [38]. Second, the weighted features are obtained using RNCA [39], which selects the high-weighted features from the original features. The importance of the BP estimation performance can be corrected by repeating the estimation experiment several times. Therefore, this study proposes an adaptive AIC of automatically calibrated likelihoods based on the GP algorithm to determine the best feature subset as a model selection problem, yielding an excellent performance with negligible computational overhead after calibration.

This study provides uncertainty for cuffless BP measurements and introduces a method for reducing the error of BP estimates based on PPG and ECG signals. As previously mentioned, the proposed study estimates the exact BPs and CIs concurrently, representing the uncertainty of cuffless BP estimation. To the best of the authors' knowledge, this study is the first to propose a GP-based feature selection and decision process (GFSDP) algorithm to simultaneously estimate cuffless BPs and CIs. Although some CI studies on conventional oscillometric BP estimation methods have been performed [5–8], studies on CI estimation in cuffless BP measurements are limited, as summarized in Table 1. The contributions of the study to BP and CI estimations are as follows:

- CIs are estimated using a bootstrap based on the GP algorithm to express uncertainty in cuffless BP estimation.
- The proposed methodology uses a hybrid F-test and RNCA to select the weighted features among the original features.
- An adaptive AIC of automatically calibrated likelihoods is proposed based on the GP algorithm to determine the best feature subset as a model selection problem.

**Table 1.** Summary of the BP estimation literature where BPs denote systolic and diastolic blood pressures, CIs are confidence intervals to represent an uncertainty [4].

| Explanation | Measurement | Goal |
| --- | --- | --- |
| Combining bootstrap with Gaussian mixture | Oscillometric | BPs, CIs [5] |
| Accuracy and "range of uncertainty" of oscillometric blood pressure monitors | Oscillometric | BPs, CIs [6] |
| BP measurement based on physiological arterial pressure variability | Oscillometric | BPs, CIs [7] |
| Estimated confidence interval from single pressure measurement | Oscillometric | BPs, CIs [9] |
| Uncertainty in blood pressure measurement | Oscillometric | BPs, CIs [10] |
| Ensemble methodology for confidence interval in oscillometric BP measurements | Oscillometric | BPs, CIs [11] |
| Confidence interval estimation for oscillometric BP measurements using bootstrap approaches | Oscillometric | BPs, CIs [8] |
| Uncertainty using the auscultatory method | Auscultatory | BPs, CIs [41] |
| Improving cuffless continuous BP Estimation with linear regression analysis | PPG, ECG | BPs [13] |
| Long short-term memory networks with transfer learning approach | PPG | BPs [12] |
| A novel neural network model for BP estimation using photoplethysmography | PPG | BPs [15] |
| A neural-network-based method for continuous BP estimation from a PPG Signal | PPG | BPs [16] |
| Joint regression network for cuffless BP estimation | PPG | BPs [14] |
| BP estimation from ECG using machine learning | ECG | BPs [20] |
| Machine learning with feature extraction for BP estimation using PPG | PPG | BPs [21] |
| A transfer learning for personalized BP estimation using PPG | PPG | BPs [22] |
| A long short-term memory and convolutional neural network using cuffless BP estimation | PPG, ECG | BPs [23] |
| Multichannel-PPG-based SVM for continuous BP estimation | PPG | BPs [19] |
| Cuffless BP monitoring system based on pulse arrival time | PPG, ECG | BPs [24] |
| Real-time cuffless continuous BP estimation using deep learning | PPG, ECG | BPs [25] |
| ECG and PPG features for cuffless blood pressure estimation using machine learning, | PPG, ECG | BPs [30] |
| Cuffless high-accuracy calibration-free BP estimation using pulse transit time | PPG, ECG | BPs [28] |
| Cuffless continuous BP estimation from pulse morphology of photoplethysmograms | PPG, ECG | BPs [27] |

## 2. Methods

Figure 1 shows the block diagram of the GFSDP algorithm. The first stage was obtained using PPG and ECG signals, as shown in Figure 1a. At this stage, a public dataset was acquired from the University of California Irvine (UCI) ML repository center [28]. Outliers were removed in the preprocessing stage. Valuable features were extracted after preprocessing as they are essential for accurate BP and CI estimations from ECG and PPG signals, as shown in Figure 1c. Subsequently, the hybrid F-test [38] and RNCA [39]

algorithms were used to select weighted features from the original features, as shown in Figure 1d. Thereafter, the adaptive AIC determined the best feature subset as a model selection problem using likelihoods based on the GP algorithm, as shown in Figure 1e. Finally, the best feature subset was used as the input feature for the GP algorithm.

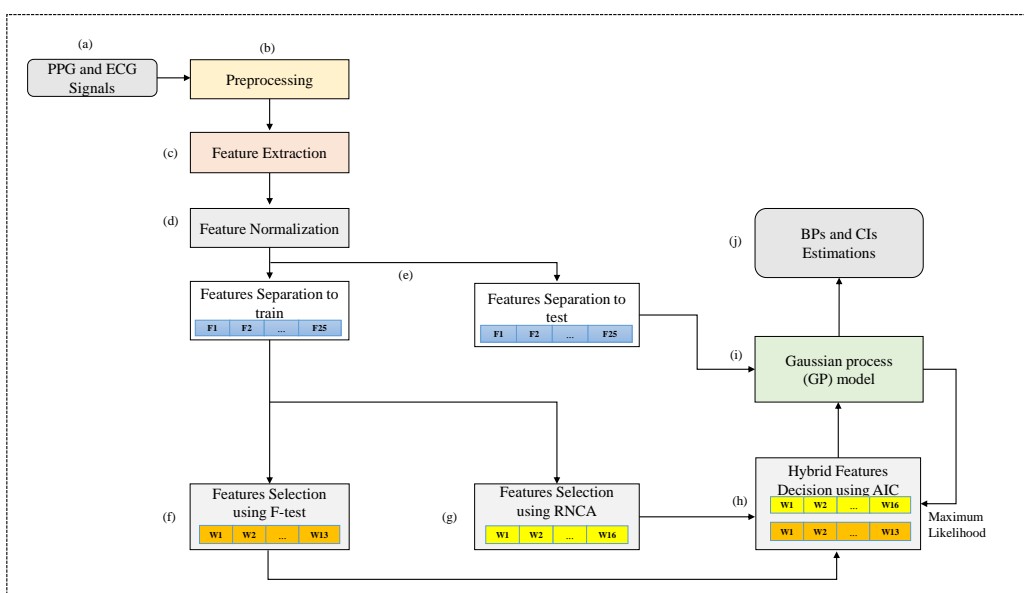

**Figure 1.** Block diagram of the proposed method using a GP-based feature selection and decision process (GFSDP) algorithm. (**a**–**j**) stand for process steps.

### 2.1. Data Set

The public dataset was collected from the UCI ML repository center [28], extracted from multiple-parameter intelligent monitoring in intensive care (MIMIC)-II data [28,42]. The database comprised ECG, finger PPG, and arterial blood pressure (ABP) signals from 3000 records at 125 Hz. The reference systolic blood pressure (SBP) and diastolic blood pressure (DBP) were calculated from the ABP signals. The feature set was obtained by combining PPG with ECG signal waveforms. Data from specific subjects with very high and very low BPs were removed according to the following conditions: (SBP $\geq$ 180, SBP $\leq$ 80, DBP $\geq$ 130, and DBP $\leq$ 50). Only the first 20 s of the record were used because each record in the database ranged from 2 to 530 s.

### 2.2. Preprocessing

Outliers were eliminated using signal processing to extract useful features from the PPG and ECG signals. First, NaNs were eliminated across all signals to preserve the alignment for each subject. PPG signals were normalized to different values for each subject using the min-max method [43]. The ECG and ABP signals were not normalized to extract only time-domain features, and the preservation of the original ABP units (mmHg) was required for estimating the SBP and DBP. The effective features were extracted from the PPG and ECG wave signals. A Kaiser window with a 35 Hz cutoff frequency and 3 dB signal bandwidth was used to eliminate high-frequency noise. Subsequently, low-frequency noise was eliminated using a Kaiser window with a 0.0665 Hz cutoff frequency and 3 dB bandwidth. Thereafter, ECG, PPG, and ABP signals were prepared into 20 s segments. The segmented signals with minimum and maximum values above or below a specific threshold were discarded. Finally, each set of 20 s window of denoised ECG, PPG, and ABP signals was segmented into smaller segments containing fewer cardiac cycles, providing an input for the feature extraction. After preprocessing, 1725 records were obtained.

## 3. Feature Selection and Decision

### 3.1. Review of Feature Extraction

Extracting valuable features after preprocessing is necessary for accurate BP and CI estimations using PPG with ECG signals [13,23,24,28]. Therefore, the time and frequency domains of the ECG and PPG signals were analyzed. However, this study considered that the frequency information was concentrated in the low-frequency band below 1.5 Hz; thus, valuable features could not be extracted in the frequency domain. Therefore, the features were extracted using the pulse morphology of the PPG signal and time between the ECG and PPG signals on the time axis, as shown in Figure 2. First, the PTT was obtained, which is the time interval between the arrival of blood flowing distally and the opening of the aortic valve [14,28,30].

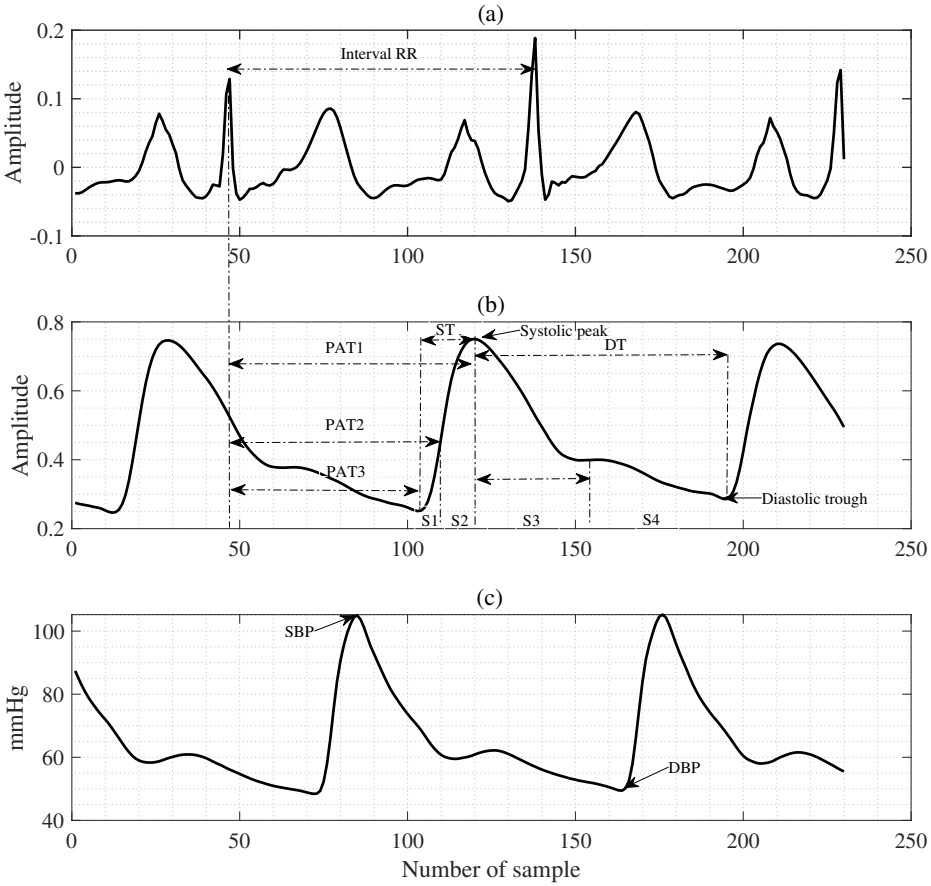

**Figure 2.** Feature extraction from the PPG with ECG signal, where (**a**) is an ECG signal example, (**b**) denotes a PPG signal example, and (**c**) is a target signal (ABP) example.

The pulse arrival time (PAT) denotes the time interval between the R peak of the ECG signal and PPG rise points, and both PAT and PTT are essential features for estimating BP values [14,24,25]. Another essential feature is the PPG's pulse intensity ratio, represented to be inversely proportional to the diastolic trough [14]. The waveform associated with the heart rate cycle can be observed through the PPG signal. Thus, the PPG signal waves were defined as pulses, each corresponding to a cardiac cycle, with the rising edge as the systolic time (ST) and falling edge as the diastolic time (DT) [14]. In addition, the area in the pulse corresponding to the ST was used as the systolic area (Sa), and the area in the pulse corresponding to the DT was used as the diastolic area (Da) [24]. As shown in Figure 2, each pulse waveform was divided into these two areas. Therefore, the features of each pulse were extracted, including Sa, Da, ST, DT, and cycle duration, to extract features that effectively estimate BPs and CIs. Figure 3 shows the 1st and 2nd derivatives of the PPGs. These results

are summarized in Tables 2 and 3. The effectiveness of the final feature set was validated and evaluated using the ANN, MLR, SVM, LSTM, GP, and proposed GFSDP algorithms.

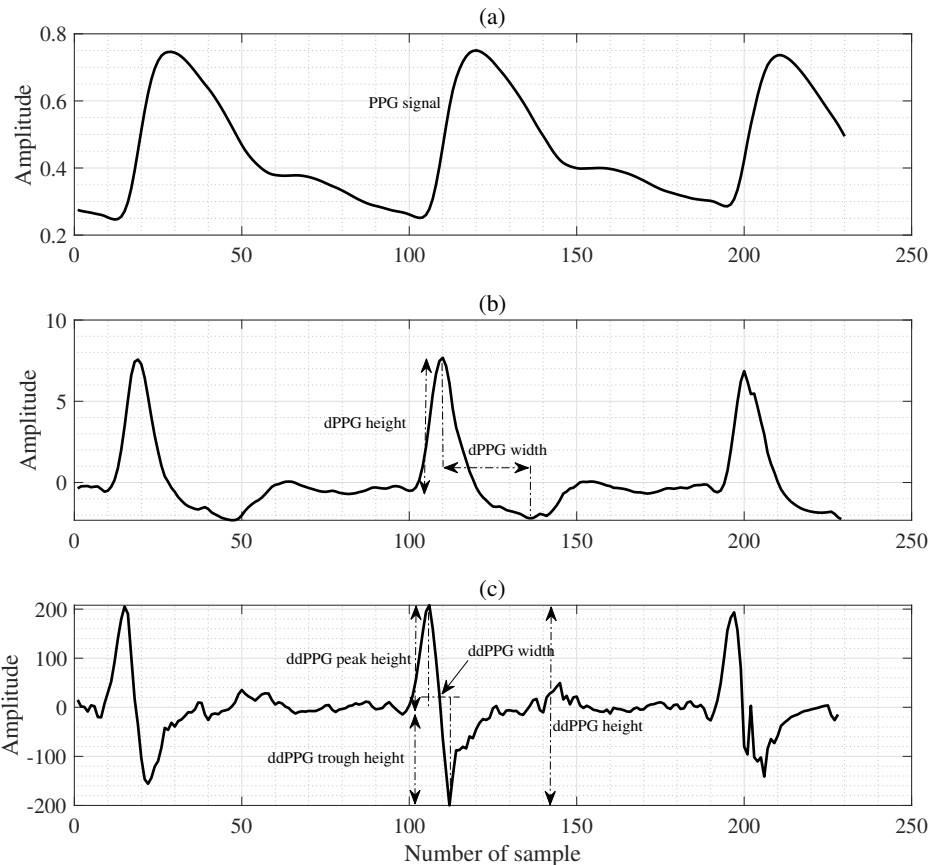

**Figure 3.** Feature extraction from the 1st and 2nd derivative of PPG waveforms, where (**a**) is a PPG signal example, (**b**) denotes the 1st derivative of the PPG wave signal, and (**c**) denotes the 2nd derivative of the PPG wave signal.

**Table 2.** Summary of the features, part I.

| Features | Explanation | Ref. |
|---|---|---|
| 1: Systolic time (ST) | Ascending time from the trough of PPG to the systolic peak | [14,26] |
| 2: Diastolic time (DT) | Descending time from the PPG's systolic peak to the next PPG morphology diastolic trough | [14,26] |
| 3: Pulse intensity ratio (PIR) | Ratio of the intensity of the PPG's systolic peak and diastolic trough | [14,26] |
| 4: Heart rate (HR) | The inverse value of the duration between consecutive ECG's R peaks | [14,26,28] |
| 5: Pulse arrival time (PAT1) | The time between the R peak of ECG and the systolic peak of PPG | [26,28] |
| 6: PAT(3) | The time between R peak of ECG and diastolic trough | [26,28] |
| 7: PAT(2) | The time between the R peak of ECG and maximum slope point (1st derivative peak value) | [26,28] |
| 8: Large artery stiffness index (LASI) | The inverse of the period from the PPG's systolic peak to the inflection point closest to the diastolic peak | [26,28] |
| 9: Augmentation index (AI) | Measure of the pressure waves reflection on arteries and it is computed through the ratio of the PPG pulse peak intensity and the intensity of the inflection point closer to the diastolic peak | [26,28] |

**Table 3.** Summary of the features, part II, where $p_m$ denotes the area under the pulse divided by the pulse duration, $p_d$ is the minimum intensity of the PPG signal, and $p_s$ denotes the maximum intensity of the PPG signal.

| Features | Explanation | Ref. |
|---|---|---|
| 10: S1 | Area under the PPG pulse curve from the diastolic trough to the point of max slope | [26,28] |
| 11: S2 | From the point of max slope to the systolic peak | [26,28] |
| 12: S3 | From the systolic peak to the inflection point closest to the diastolic peak | |
| 13: S4 | From the inflection point to next pulse's diastolic trough | [28] |
| 14: Inflection point area ratio (IPAR) | Ratio of S4/(S1+S2+S3) | [26,28] |
| 15: PPG$_k$ | $(p_m - pd)/(p_s - pd)$ | [26] |
| 16: dPPG height (H) | PPG's 1st derivative characteristics | [14,28] |
| 17: dPPG width (W) | PPG's 1st derivative characteristics | [14,26] |
| 18: ddPPG peak height (PH) | PPG's 2nd derivative characteristics | [14,26] |
| 19: ddPPG trough height (TH) | PPG's 2nd derivative characteristics | [14,26] |
| 20: ddPPG width (W) | PPG's 2nd derivative characteristics | [14,26] |
| 21: ddPPG height (H) | PPG's 2nd derivative characteristics | [14,26] |
| 22: MXAP | The pulse's maximum amplitude | [30] |
| 23: MIAP | The pulse's minimum amplitude | [30] |
| 24: MEU | The blood's viscosity | [30] |
| 25: FHR | The frequency of HR | [30] |

*3.2. Hybrid Feature Decision*

This study proposes a hybrid feature selection algorithm that integrates the F-test, RNCA, and AIC to select valuable features for BP and CI estimations. Feature selection typically comprises two steps: calculating the weight for each feature and selecting the optimal subset as input features. Therefore, the F-test [38] was used to obtain a set of ranked features. Subsequently, the RNCA [39] was applied to select weighted feature vectors from among the original features. The AIC [40] was then used to determine the best feature subset. Finally, the AIC measured the quality of the selected features from the F-test and RNCA to determine the hybrid situation, as indicated in Algorithm 1. Feature selection involves selecting essential features that are more uniform, nonoverlapping, and effective in improving the performance of ML algorithms. As mentioned in the introduction, the best feature subset with high weights was obtained from the original feature set. Additionally, reducing the number of features by determining which features have higher weights makes ML algorithms easier to understand and explain.

Feature Weighting Using the F-Test

This study used the F-test to select meaningful features for BP estimation [38]. The F-test is a statistical test weighted by computing the variance ratio. In this study, the F-test based on a one-way ANOVA was used to calculate the between- and within-group variance ratios for each feature. A group represented the instances with the same response value. Higher weights indicated shorter intragroup distances and greater intergroup distances. Thus, features were ranked based on higher weights using an F-test based on a one-way ANOVA [38]. The null hypothesis was that the target values grouped by function in each F-test were drawn from populations with the same mean, as opposed to the alternative hypothesis that all population means were unequal:

$$\mathcal{H}_0 : \mu_1 = \mu_2 = ... = \mu_m, \quad \mathcal{H}_1 : \mu_1 \neq \mu_2 \neq ... \neq \mu_m \tag{1}$$

where $m$ is the number of group and $\mu_m$ denotes the mean for group $m$. The overall mean was calculated as

$$\mu = \frac{1}{n} \sum_{k=1}^{m} \mu_k n_k, \quad (n = \sum_{k=1}^{m} n_k) \tag{2}$$

where $n_k$ is the number of the $k$th group. Hence, the mean of a real feature's samples was given by

$$\bar{x}_k = \frac{1}{n} \sum_{i=1}^{n_k} x_{ik}, \tag{3}$$

The total mean was computed as

$$\bar{x} = \frac{1}{n} \sum_{k=1}^{m} \sum_{i=1}^{n_k} x_{ik}, \tag{4}$$

The sum of the mean squared deviation (MSD) within groups was given as

$$\mathcal{S}_W = \frac{1}{n} \sum_{k=1}^{m} \sum_{i=1}^{n_k} (x_{ik} - \bar{x}_k)^2 \tag{5}$$

The sum of the MSD between groups was computed by

$$\mathcal{S}_B = \frac{1}{n} \sum_{k=1}^{m} \sum_{i=1}^{n_k} (\bar{x}_k - \bar{x})^2 \tag{6}$$

Hence, we obtained the F-score as

$$\mathcal{F}_S = \frac{\mathcal{S}_B / (m-1)}{\mathcal{S}_W / (n-m)} \tag{7}$$

---

**Algorithm 1** Hybrid F-test and RNCA using AIC based on GP algorithm.

---

1:  **Procedure** F-TEST(**X**, **Y**): training data set
2:  return $(w_f)$ that produces weighted feature vectors using F-test
3:  select $(w_f) \geq$ threshold
4:  **EndProcedure**
5:  **Procedure** RNCA(**X**, **Y**): divided data set into 5 folds
6:  **for** $i = 1, n$ **do**
7:      $\lambda_{i,k}$: tuning using 5-fold cross-validation
8:      **for** $k = 1, 5$ **do**
9:          call NCA($D, \lambda_{i,k}$): train NCA for $\lambda$
10:          compute $\mathcal{L}_{i,k}$: record loss values
11:      **end for**
12: **end for**
13: $\mathcal{C}_\mu = $ mean$(\mathcal{C}_{k,i})$: calculate average costvalue
14: $\lambda_b = \arg\min_{\mathcal{C}_\mu} (y|x, \lambda_{k,i}, \mathcal{C}_\mu)$: detect best $\lambda_b$
15: call RNCA($\mathcal{D}, \lambda, \zeta$): $\zeta = @(y_k, y_j) 1 - \exp(-|y_k - y_j|)$
16: return $(w_r)$ weighted features using RNCA
17: select $(w_r) \geq$ threshold
18: **EndProcedure**
19: **Procedure** AIC
20: call GP (**X**, **Y**): return $\mathcal{M}$ a maximum likelihood
21: call AIC $(\mathcal{M}, \mathcal{W})$ : $\mathcal{W}$ is the number of $(w_f, w_r)$
22: decide $(w_b)$: the best feature subset
23: **EndProcedure**

---

The F-test accepts an alternative hypothesis if $p < 0.05$. This indicates a significant difference in a feature between the two groups. When $p \geq 0.05$, the null hypothesis is accepted, and the alternative hypothesis is rejected. This indicates that there is no significant difference in a feature between the two groups. The smaller the $p$-value, the more significant the difference in a feature between the two groups, making it more beneficial for estimating BPs. Therefore, a small $p$-value for the test statistic indicates the importance of a feature, as shown in Figure 4a,b. Table 4 lists the ranked features obtained using the F-test.

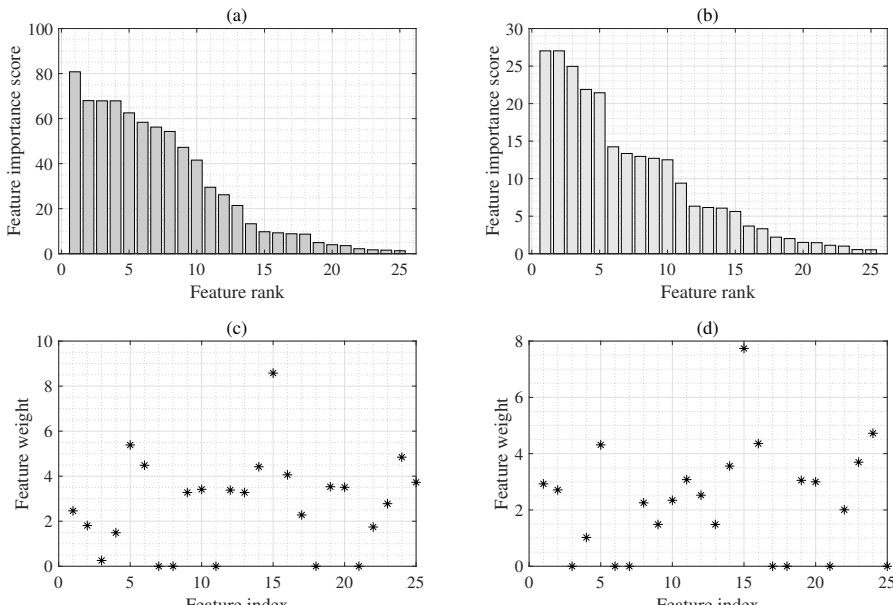

**Figure 4.** The weighted feature sets were obtained using the F-test and RNCA algorithms, where (**a**,**b**) denote the ranked features acquired using the F-test and (**c**,**d**) were obtained from the RNCA.

**Table 4.** The high-score ranked features were selected using the F-test and RNCA algorithms for the SBP and DBP estimations.

| | F-test | | RNCA | | | F-test | | RNCA | |
|------|--------|--------|--------|--------|------|--------|--------|--------|--------|
| Rank | SBP | DBP | SBP | DBP | Rank | SBP | DBP | SBP | DBP |
| 1 | dppgH | HR | HR | HR | 14 | S2 | S4 | PPGk | MXAP |
| 2 | ddppgPH | PAT1 | ST | ST | 15 | PPGk | PIR | S4 | IPA |
| 3 | PAT2 | dppgH | DT | ddppgH | 16 | MXAP | MEU | FHR | MIAP |
| 4 | ddppgH | ddppgPH | PAT3 | AI | 17 | DT | IPA | MXAP | MEU |
| 5 | PAT1 | PAT3 | ddppgW | PAT1 | 18 | MEU | DT | MEU | DT |
| 6 | ddppgFH | ST | ddppgFH | PAT3 | 19 | LASI | dppgW | IL | dppgH |
| 7 | ST | ddppgH | AI | S2 | 20 | AI | FHR | IPA | ddppgPH |
| 8 | PAT3 | LASI | PAT1 | ddppgW | 21 | PIR | ddppgW | dppgW | S4 |
| 9 | HR | PAT2 | S3 | ddppgFH | 22 | FHR | MXAP | PAT2 | S1 |
| 10 | S3 | S3 | S2 | FHR | 23 | S4 | S1 | LASI | LASI |
| 11 | dppgW | ddppgFH | ddppgH | dppgW | 24 | IPA | MIAP | S1 | PIR |
| 12 | ddppgW | S2 | dppgH | S3 | 25 | MIAP | AI | PIR | PAT2 |
| 13 | S1 | PPGk | ddppgPH | PPGk | | | | | |

### 3.3. Feature Selection Using Neighbor Component Analysis (NCA)

Feature selection involves selecting essential features from the original feature set. This implies that only a few features affect the target BPs. Thus, reducing the dimensions of the feature space while retaining only valid information is essential for BP estimation.

Weighted feature vectors were extracted from the original feature set using the RNCA algorithm [39]. The NCA method [44] trains a weighted feature vector by minimizing a loss function, in which diagonal adaptation measures the mean deviation leave-one-out regression loss from the training dataset. Thus, a dataset is defined as

$$D_w = \sum_{m=1}^{p} w_m^2 |x_{im} - x_{jm}|, \tag{8}$$

Here, $D_w$ is the weighted distance and $w_m$ denotes the $m$th weighted feature. Thus, the probability $P(\gamma(x) = x_j|\mathcal{T}_d)$ that point $x$ is chosen from $\mathcal{T}_d$ as the reference point is:

$$P(\gamma(x) = x_j|\mathcal{T}_d) = \frac{\mathrm{k}(D_w(x_i - x_j))}{\sum_{j=1}^n \mathrm{k}(D_w(x_i - x_j))} \tag{9}$$

Here, $(\mathrm{k}(z) = \exp(-z/\sigma))$ denotes the kernel, and the kernel width $\sigma$ is a parameter that affects the probability that each point is chosen as a reference [44]. Here, we assumed that $P(\gamma(x) = x_j|\mathcal{T}_d) \propto \mathrm{k}(D_w(x_i, x_j))$ and estimated the response to $x_i$ using the training data set in $\mathcal{T}_d^{-i}$, $(x_i, y_i)$. The probability that $x_j$ is chosen as the reference point for $x_i$ is given as

$$\gamma_{ij=}P(\gamma(x) = x_j|\mathcal{T}_d^{-i}) = \frac{\mathrm{k}(D_w(x_i - x_j))}{\sum_{j=1, j\neq i}^n \mathrm{k}(D_w(x_i - x_j))} \tag{10}$$

$$\mathcal{L}_i = \mathbf{E}(\mathcal{L}(y_i, \hat{y}_i)|\mathcal{T}_d^{-i}) = \sum_{j=1, j\neq i}^n \gamma_{ij}\mathcal{L}(y_i, y_j) \tag{11}$$

where $\mathcal{L}$ is the loss function that gives the difference between $(\hat{y}_i, y_i)$. Thus, we applied the regularization parameter $\lambda$ to minimize the loss function as follows:

$$\mathcal{F}_w = \frac{1}{n}\sum_{i=1}^n \mathcal{L}_i + \lambda \sum_{m=1}^p w_m^2 \tag{12}$$

Hence, we used the regularization parameter to choose weighted feature vectors from high-dimensional features employing the NCA algorithm as given in (8)–(12), with $\lambda(= 0.015)$ [44] .

### 3.4. RNCA

The performance of the RNCA algorithm is affected by the normalization parameter $\lambda$. Therefore, the RNCA algorithm must be defined to set the parameters effectively. The regularization parameter was adapted using the mean squared error and 5-fold cross-validation, as shown in (Algorithm: RNCA). A user-defined robust loss function was applied, given as $\zeta = 1 - \exp(-|y_i - y_j|)$. Thus, the value of $\lambda$ was determined to represent the minimum average loss value. Finally, the weighted feature vectors were detected using RNCA without selecting other features, as shown in Table 4. Therefore, the feature selection reduced the dimensionality of the algorithm training.

AIC Based on GP Algorithm

The AIC can be used to compare and rank multiple feature sets and determine the most appropriate feature set [40]. After computing several other groups, they were compared using this criterion. A feature set was considered as a group. According to Akaike's theory, the best feature subset has the lowest AIC value.

$$\mathcal{A} = -2\ln(\mathcal{M}) + 2\eta \tag{13}$$

where $\mathcal{M}$ denotes the maximum likelihood obtained from the previous result of the GP algorithm, and $\eta$ represents the number of feature dimensions selected. The weight of the features can change according to the target variables SBP and DBP, as shown in Table 4. The weight of the features also changes according to repeated experiments. Thus, an effective feature subset that could most accurately estimate SBP and DBP was determined using the proposed GFSDP algorithm.

### 3.5. GP Based on Bayesian Inference

This section describes the GP regression [31] used to train and test the proposed algorithm. Owing to the size of the paper, the description of the conventional algorithms is

omitted. The GP algorithm is a robust, flexible, nonparametric Bayesian algorithm used in supervised ML [31]. To train the GP algorithm, the explanatory and response variables should be prepared as the input and output data, respectively, as $\mathbf{D} = \{x_i, y_i\}_{i=1}^{I}$, $x \in \mathbf{R}^{I \times D}$, and $y \in \mathbf{R}^{I \times 1}$. Here, we used a mapping function $f_m = f(x)$ to estimate $y$ given $x$. Hence, we assumed that the response variable $y$ was acquired using the corresponding $x^{\mathrm{T}} w$ by including noise, as follows

$$y = x^{\mathrm{T}} w + \varepsilon, \quad \varepsilon \backsim \mathbf{N}(0, \sigma^2 \mathrm{I}) \tag{14}$$

The weighted vectors $w$ and variance $\sigma^2$ were acquired from the resampled signal dataset. The GP algorithm estimates the response variable based on Gaussian processes (GPs) using the mapping function $f_m(x)$ and explicit essential functions $\beta$.

$$f_m(x) \backsim \mathbf{GP}(0, \mathrm{k}(x, x')) \tag{15}$$

where $f_m(x)$ is acquired from a zero-mean GP algorithm using a covariance function $\mathrm{k}(x, x')$ [32]. Hence, we can obtain the mapping function $f_m(x) = \beta(x)^{\mathrm{T}} w$. The mean function of the input data can be defined as the expected value of the mapping function $\theta(x) = \mathbf{E}[f_m(x)]$. A latent variable covariance function obtains the smoothness of the response variables, and the basic function projects the input data $x$ into the dimensional feature space.

$$\mathrm{k}(x, x') = \mathbf{E}[(f_m(x) - \theta(x))(f_m(x') - \theta(x'))^{\mathrm{T}}] \tag{16}$$

We defined the expected value of (16) as

$$\mathrm{k}(x, x'|\eta) \approx \sigma^2 \exp\left(-\frac{\|x - x'\|^2}{2\eta^2}\right) \tag{17}$$

where k is a kernel for the GP [31], $\eta$ is a hyperparameter, and $\sigma^2$ is a variance based on resampled signals. In the study, we used exponential squares as the kernel, as in (17). Thus, the kernel decided the properties of the mapping function $f_m(x)$. We could define an instance of response variables **y** using the Bayesian inference based GP as

$$p(y_i | f_m(x_i), x_i) \backsim \mathbf{N}\big(y_i | \beta(x_i)^{\mathrm{T}} w + f_m(x_i), \sigma^2\big) \tag{18}$$

where $\beta(x_i)$ denotes a basic function transforming the original explanatory variable $x$ into a new variable $\beta(x)$. Thus, we determined $\mathbf{\Theta} = \{w, \eta, \sigma^2\}$ from the dataset **D**, and the marginal likelihood was expressed as

$$p(y|x) = p(y|x, \mathbf{\Theta}) \approx \mathbf{N}(y|\mathbf{\Omega} w, \mathrm{k}(x, x'|\eta) + \sigma^2 \mathrm{I}), \tag{19}$$

Generally, the local maxima for the hyperparameter $\mathbf{\Theta}$ can be determined and used to train the GP algorithm. In addition, choosing an appropriate kernel depends on hypotheses, such as the smoothness and expected patterns of the data. By maximizing the log marginal likelihood, we could estimate the hyperparameter $\mathbf{\Theta}$, as follows

$$\log p(y|x, \mathbf{\Theta}) = -\frac{1}{2} \log \left|\mathrm{k}(x, x'|\eta) + \sigma^2 \mathrm{I}\right| - \frac{1}{2} i \log 2\pi$$
$$- \frac{1}{2}(y - \mathbf{\Omega} w)^{\mathrm{T}} \big[\mathrm{k}(x, x'|\eta) + \sigma^2 \mathrm{I}\big]^{-1}(y - \mathbf{\Omega} w) \tag{20}$$

where $\mathrm{k}(x, x'|\eta)$ is the kernel matrix and $\mathbf{\Omega}$ denotes the matrix of the explicit basic function. Herein, we applied a penalty-fitting scale to represent the logarithmic likelihood and maxi-

mized it using a gradient approach using optimization techniques. The hyperparameters $\Theta = \{w, \eta, \sigma^2\}$ using the GP algorithm maximized the likelihood $p(y|x)$ as a function of $\Theta$.

$$\mathcal{L}(\widehat{\Theta}) = \arg\max_{\Theta} \log(y|x, \Theta) \tag{21}$$

First, we determined $\hat{w}(\eta, \sigma^2)$ to predict hyperparameters that maximized the log-likelihood concerning $w$ for a given $(\eta, \sigma^2)$ as

$$\hat{w}(\eta, \sigma^2) \quad = \quad \left\{\Omega^{\mathrm{T}}\left[k(x, x'|\eta) + \sigma^2 I\right]^{-1}\Omega\right\}^{-1}\Omega^{\mathrm{T}}\left[k(x, x'|\eta) + \sigma^2 I\right]^{-1}y \tag{22}$$

Second, we needed a probability density function $p(\mathbf{y}^*|\mathbf{y}, \mathbf{x}, \mathbf{x}^*)$ for the probabilistic estimation of the Bayesian GP algorithm using known hyperparameters. However, we estimated a response variable $y$ using a finite amount of new input data $x^*$ and predicted the output of these data based on a multivariate Gaussian distribution with a kernel-generated covariance matrix. Thus, we denoted the conditional probability distribution, as follows.

$$p(y^*|y, x, x^*) = \frac{p(y^*, y|x, x^*)}{p(y|x, x^*)} \tag{23}$$

In order to acquire the joint density probability function in the numerator, as expressed in (23), the mapping functions $f_m^*$ and $f_m$ should be used, as follows.

$$p(y^*, y|x, x^*) = \int\int p(y^*, y, f_m^*, f_m|x, x^*)df df^*$$
$$= \int\int p(y^*, y|f_m^*, f_m, x, x^*)p(f_m^*, f_m|x, x^*)df df^* \tag{24}$$

The GP algorithm assumes that each response variable $y_i$ depends only on the corresponding latent variable $f_m(x_i)$ and input vector $x_i$. Given $y, x$ and the hyperparameters $\Theta$, the expected value of the estimation is given as:

$$\mathbf{E}(y^*|y, x, x^*, \Theta) = \theta(\mathbf{x}^*)^{\mathrm{T}}\mathbf{w} + \mathbf{c}(\mathbf{x}, \mathbf{x}'|\eta)\varphi$$
$$= \beta(x^*)^{\mathrm{T}}w + \sum_{i=1}^{I}\varphi_i k(x^*, x_i|\eta) \tag{25}$$

where $\varphi = [k(x, x) + \sigma^2 I]^{-1}(y - \Omega w)$. Practically, we determined an optimal point prediction $\widehat{y}^*$ based on the loss function as

$$\mathbf{E}_{\mathcal{L}}(\widehat{y}^*|x^*) = \int \mathcal{L}(y^*, \widehat{y}^*)p(y^*|x^*, \mathbf{D})dy^* \tag{26}$$

We obtained an predicted $y^* \approx \widehat{y}^*$ and minimized the expected value of the loss function $\mathcal{L}(y^*, \widehat{y}^*)$ by minimizing between $y^*$ and $\widehat{y}^*$ as

$$\hat{y}_{\mathrm{opt}}|x^* = \arg\min_{\widehat{y}^*} \mathbf{E}_{\mathcal{L}}(\widehat{y}^*|x^*) \tag{27}$$

In this study, we used the mean absolute error (MAE) as the loss function, given by: $\mathcal{L}$. Generally, we can use the MAE metrics to evaluate the estimation accuracy.

### 3.6. CIs Estimation

CIs were calculated using the bootstrap method to represent the uncertainty of the estimated BPs (SBP and DBP) for the conventional and proposed algorithms. This was denoted as the bootstrap concept of the parametric algorithm. Each patient had four to eight estimated BP results, and many pseudo-BPs could be generated using these estimated results, $\mathbb{E} = [e_1, \ldots, e_n]$, using $n$ independent measurements from an unknown distribution $\mathbb{T}$ to estimate a CI for $\widehat{\mu}(\mathbb{E})$.

In addition, we assumed that $\mathbb{E}$ was the random data of the distribution $\mathbb{T}$ with unknown parameter $[\mu, \sigma]$ to estimate a CI for $\hat{\mu}(\mathbb{E}^*)$. Thus, we obtained pseudo-BP results $\mathbb{E}^* = [e_1^*, \ldots, e_n^*]$ from $\widehat{\mathbb{T}}(\widehat{\mu}, \widehat{\sigma} | \mathbb{E})$ based on the Monte Carlo algorithm, where $[\widehat{\mu}, \widehat{\sigma}]$ commonly denotes a maximum likelihood estimate from $\mathbb{E} = [e_1, \ldots, e_n]$. In this study, we obtained the CIs using the bootstrap technique [10,35], which could be acquired using the estimated BP results with respect to each algorithm. Subsequently, we obtained a matrix as:

$$\mathbf{M}^*(i \mid \widehat{\mathbb{E}}_i^*) = \begin{bmatrix} e_{1,1}^{*i} & \cdots & e_{1,B}^{*i} \\ \vdots & \ddots & \vdots \\ e_{n,1}^{*i} & \cdots & e_{n,B}^{*i} \end{bmatrix} \tag{28}$$

where (28) was obtained as $\widehat{\mu}_i^* + \widehat{\sigma}_i^* \times \mathrm{rand}(n, B)$, then we calculated vertically each column to acquire the average of each column as $\widehat{\mu}_b^* = 1/n \sum_{j=1}^{n} e_{j,b}^{*i}$, where$*$ denotes the resampled BPs acquired from the bootstrap algorithm. Then, we performed ascending sorts and the sorted BP results were given by $(\widehat{\mu}_1^*, \widehat{\mu}_2^*, \cdots, \widehat{\mu}_B^*)$, supposing $\widehat{\mu}_\alpha^*$ denotes the $100\alpha$th percentile of $B(=100)$ bootstrap replications $(\widehat{\mu}_1^*, \widehat{\mu}_2^*, \cdots, \widehat{\mu}_B^*)$. We could obtain the CI as $\widehat{\mu}_{\mathrm{lower}}^*, \widehat{\mu}_{\mathrm{upper}}^*$ of the $1 - 2 \cdot \alpha$, from this bootstrap algorithm as $(\widehat{\mu}_\alpha^*, \widehat{\mu}_{1-\alpha}^*)$.

## 4. Experimental Results

### 4.1. Parameter Fine-Tuning and Complexity of the ML Model

The parameters of the proposed GFSDP and conventional algorithms were set before training. These parameters are essential because they can improve the algorithm's performance when configured effectively. This study applied a fivefold cross-validation and grid search to fine-tune these parameters. The core parameters for each ML algorithm were defined, and the range of possible values for each parameter was determined.

Subsequently, a grid search on all possible combinations of the parameters was performed for each ML algorithm, which observed the best parameter sets that helped the ML algorithms obtain the highest results. Finally, a fivefold cross-validation was performed to improve the robustness of the algorithm with the optimal parameters obtained. The training data were randomly segregated into five nonoverlapping subsets of equal sizes. Five iterations were obtained: four folds were used for learning each iteration, and the remaining fold was applied to perform the evaluation. The final output was the average of the five folds. Table 5 lists the parameter ranges for each parameter of each ML algorithm under consideration and the optimized parameters after performing the grid search. The feature training and testing times were calculated using MATLAB ®2022 [45], as listed in Table 6. The proposed GFSDP algorithm required more computational time than the GP algorithm using the MIMIC II dataset.

### 4.2. Evaluation Protocols

The selected feature subset was then randomly split into 80% for training and 20% for testing. Subsequently, systolic reference BPs and diastolic BPs were calculated from the ABP envelope signals, as shown in Figure 2c. First, the mean absolute error (MAE) and standard deviation of the error (SDE ) of the GFSDP algorithm were compared with those of the conventional algorithms to evaluate the experimental results, as shown in Table 7. The probability of the British Hypertension Society (BHS) protocol [46] was also obtained based on the results of the MAE and SDE, as shown in Table 8. The average error of the GFSDP algorithm was calculated by $er_i = (ep_i - rp_i)$ for each record $i$, where $ep$ denotes the estimated BPs (SBP or DBP), and $rp$ is a reference BP. Thus, the mean error (ME) and MAE were given as $(\frac{1}{n} \sum_{i=1}^{n} er_i)$ and $(\frac{1}{n} \sum_{i=1}^{n} |er_i|)$, respectively. The results were obtained as the average of the 30 experiments for each algorithm.

Second, the ME and SDE of the ME were used [10], as shown in Table 9. The ME and SDE between the estimated and calculated reference BPs were calculated using the recommendations of the Association for the Advancement of Medical Instrumentation (AAMI) protocol [47]. A device passes the AAMI protocol if its measurement error has an

ME value of less than 5 mmHg with an SDE of less than 8 mmHg [47]. Table 10 lists the CI results of the cuffless BP signals. The CIs obtained using the proposed algorithm were lower than those obtained using the conventional algorithm for acquiring CIs using an SVM with a Student's t-distribution (SVM$_{ST}$) [8].

**Table 5.** Summary of the parameters of the proposed GFSDP and conventional algorithms, where SE is a squared exponential kernel for the GP algorithm.

| Parameters | ANN | MLR | SVM | LSTM | GP | GFSDP |
|---|---|---|---|---|---|---|
| Input dimension | 25 | 25 | 25 | 25 | 25 | 13 to 16 |
| Output dimension | 1 | 1 | 1 | 1 | 1 | 1 |
| Optimizer | Bayes | Bayes | Bayes | Bayes | Bayes | Bayes |
| Activation | Relu | - | - | Relu | - | - |
| Hidden unit on the layers | - | - | | 200 to 300 | - | - |
| Iterations | - | - | - | 1000 | - | - |
| Fully connected layer | - | - | - | 50 | - | - |
| Dropout | - | - | - | 50% | - | - |
| Max epoch | - | - | - | 200 | - | - |
| Gradient threshold | - | - | - | 1 | - | - |
| Epsilon | - | - | 0.5 | 1.00 e-08 | - | - |
| Weight threshold | - | - | - | - | - | 1 to 3 |
| Shrinkage factor | - | - | 0.05–0.1 | - | - | - |
| Subsampling factor | - | - | 0.1–0.5 | - | - | - |
| Kernel function | - | - | Gauss. | - | SE | SE |

**Table 6.** Compared feature training and testing times between the proposed and conventional methods based on hardware (Intel®Core(TM) i5-9400 CPU 4.1 GHz, OS 64 bit, RAM 16.0 GB), and software (Matlab®2022 (The MathWorks Inc., Natick, MA, USA)) specifications.

| Algorithm | ANN | MLR | SVM | LSTM | GP | GFSDP |
|---|---|---|---|---|---|---|
| Time (s) | 1.67 | 0.13 | 0.37 | 7.48 | 3.84 | 5.56 |

**Table 7.** MAE (SDE) relative to the reference ABP and the conventional ANN, MLR, SVM, LSTM, GP and using the proposed GFSDP algorithms.

| mmHg | ANN | | MLR | | SVM | | LSTM | | GP | | GFSDP | |
|---|---|---|---|---|---|---|---|---|---|---|---|---|
| | SBP | DBP | SBP | DBP | SBP | DBP | SBP | DBP | SBP | DBP | SBP | DBP |
| MAE | 12.23 | 8.33 | 12.49 | 7.67 | 11.37 | 7.23 | 10.78 | 6.84 | 10.03 | 6.70 | 7.66 | 5.47 |
| SDE | 0.61 | 0.78 | 0.54 | 0.57 | 0.50 | 0.55 | 0.98 | 0.30 | 0.49 | 0.37 | 0.23 | 0.56 |

**Table 8.** We used the results of the ANN, MLR, SVM, LSTM, GP algorithms and the proposed GFSDP algorithm to grade the algorithm based on the BHS standard [46], where each result represents the average of 30 experimental data.

| Methods | SBP Mean Absolute Difference (%) | | | DBP Mean Absolute Difference (%) | | | SBP/DBP BHS Grade |
|---|---|---|---|---|---|---|---|
| | ≤5 mmHg | ≤10 mmHg | ≤15 mmHg | ≤5 mmHg | ≤10 mmHg | ≤15 mmHg | |
| ANN | 26.75 | 49.65 | 68.35 | 39.35 | 72.39 | 88.97 | C/C |
| MLR | 29.45 | 49.04 | 65.94 | 47.89 | 79.71 | 87.42 | C/C |
| SVM | 30.64 | 54.47 | 70.96 | 54.61 | 79.48 | 88.13 | C/C |
| LSTM | 34.20 | 60.07 | 73.16 | 48.65 | 79.48 | 90.65 | C/C |
| GP | 36.30 | 62.07 | 76.56 | 48.65 | 80.71 | 90.71 | C/C |
| GFSDP | 49.49 | 73.01 | 84.97 | 60.26 | 85.39 | 93.74 | C/B |
| Grade A | 60 | 85 | 95 | 60 | 85 | 95 | [46] |
| Grade B | 50 | 75 | 90 | 50 | 75 | 90 | |
| Grade C | 40 | 65 | 85 | 40 | 65 | 85 | |

**Table 9.** ME and SDE relative to the reference ABP and the conventional ANN, MLR, SVM, LSTM, GP algorithms and using the proposed GFSDP algorithm.

| mmHg | ANN | | MLR | | SVM | | LSTM | | GP | | GFSDP | |
| | SBP | DBP | SBP | DBP | SBP | DBP | SBP | DBP | SBP | DBP | SBP | DBP |
|------|-----|-----|-----|-----|-----|-----|------|------|----|-----|-------|-----|
| ME | 0.17 | 0.27 | 0.57 | −2.81 | 0.46 | −2.59 | −1.29 | −1.46 | 0.10 | 0.10 | 0.08 | 0.15 |
| SDE | 15.68 | 12.75 | 15.93 | 11.21 | 14.70 | 10.96 | 14.22 | 9.87 | 13.30 | 9.28 | 10.83 | 8.31 |

**Table 10.** Summary of the CIs for SBP and DBP compared to the proposed and conventional methods, where ST denotes the Student's t-distribution and BT is the bootstrap method.

| Methods | SBP (SDE) 95% CI | Lower CIs | Upper CIs | DBP (SDE) 95% CI | Lower CIs | Upper CIs |
|---------|------------------|-----------|-----------|------------------|-----------|-----------|
| $ANN_{ST}$ | 15.81 (10.23) | 114.39 (11.35) | 130.21 (11.82) | 10.96 (6.73) | 65.88 (4.05) | 76.76 (5.12) |
| $MLR_{ST}$ | 18.54 (10.30) | 113.37 (10.66) | 131.91 (10.99) | 7.31 (4.80) | 66.08 (1.51) | 73.38 (3.60) |
| $SVM_{ST}$ | 20.83 (9.83) | 112.16 (8.55) | 132.99 (10.36) | 6.05 (3.54) | 66.26 (2.80) | 72.32 (2.84) |
| $LSTM_{ST}$ | 18.61 (10.13) | 113.20 (9.68) | 131.81 (9.83) | 6.48 (4.10) | 65.88 (3.25) | 72.36 (3.40) |
| GP | 21.38 (3.56) | 120.92 (10.43) | 142.30 (8.03) | 13.74 (1.19) | 64.83 (2.85) | 78.57 (3.17) |
| GFSDP | 20.09 (3.27) | 123.49 (15.59) | 143.58 (13.08) | 13.98 (1.33) | 64.42 (1.97) | 78.40 (2.25) |
| $GFSDP_{BT}$ | 11.19 (8.59) | 118.34 (7.77) | 129.53 (10.07) | 2.19 (2.47) | 70.75 (1.34) | 75.94 (2.28) |

*4.3. Statistical Results*

This study used the ANN, MLR, SVM, LSTM, and GP algorithms [17,31] as conventional algorithms to evaluate the performance of the proposed GFSDP algorithm. The first experiment obtained the objective MAE and SDE results. The MAEs of the SBP (12.23 mmHg) and DBP (8.33 mmHg) obtained using the ANN algorithm were compared with the reference BP shown in Table 7. The MAEs of the MLR algorithm for the SBP (12.49 mmHg) and DBP (7.67 mmHg) were compared with the reference BP. Table 7 shows the MAEs of the SBP (11.37 mmHg) and DBP (7.23 mmHg) obtained using the SVM algorithm. The MAEs of LSTM algorithm for the SBP (10.78 mmHg) and DBP (6.98 mmHg) were compared with the reference BP. Finally, the MAEs of the SBP (10.03 mmHg) and DBP (6.70 mmHg) acquired using the GP algorithm are shown. As shown in Table 7, the proposed GFSDP algorithm obtained lower MAEs for the SBP (7.66 mmHg) and DBP (5.47 mmHg) compared with the conventional ANN, MLR, SVM, LSTM, and GP algorithms.

The GFSDP algorithm was compared with the conventional ANN, MLR, SVM, LSTM, and GP algorithms, according to the British hypertension protocol (BHS) [46]. The MAEs for the three groups of less than 5 mmHg, less than 10 mmHg, and 15 mmHg were evaluated. Table 8 shows the BHS grading obtained using the GFSDP algorithm: grades C and B were obtained for the SBP and DBP, respectively. The readings obtained using the proposed method in the test scenario were 49.49% (≤5 mm Hg), 73.01% (≤10 mmHg), and 84.97% (≤15 mmHg) for the SBP and 60.26 (≤5 mmHg), 85.39 (≤10 mmHg), and 93.74 (≤15 mmHg) for the DBP. The probabilities of the proposed algorithm based on the BHS were higher than those obtained using the conventional ANN, MLR, SVM, LSTM, and GP algorithms, as shown in Table 8.

Table 9 shows the SDEs of the MEs for the SBP (15.68 mmHg) and DBP (12.75 mmHg) obtained using the ANN algorithm compared with the reference BP. In addition, the SDEs of the MEs for the SBP (15.93 mmHg) and DBP (11.21 mmHg) from the MLR algorithm, for the SBP (14.70 mmHg) and DBP (10.96 mmHg) obtained using the SVM algorithm, and for the SBP (14.22 mmHg) and DBP (9.87 mmHg) from the LSTM algorithms are also presented. Table 9 shows the SDEs of the MEs for the SBP (13.30 mmHg) and DBP (9.28 mmHg) obtained using the GP algorithm. The GFSDP algorithm obtained lower SDEs of the MEs for the SBP (10.83 mmHg) and DBP (8.31 mmHg) than the conventional ANN, MLR, SVM, and GP algorithms. In addition, the Bland–Altman method [48] comparing the proposed GFSDP algorithm with the reference ABP is shown in Figure 5. The agreement between the conventional SVM and ABP reference values was also compared by Bland–Altman

plots (Figure 5c,d). The limits of agreement (bold horizontal lines in Figure 5) used in this study were (ME $\pm$ 2 $\times$ SD) for all plots. The bias (horizontal center lines) for all plots was small ($\leq\pm0.5$ mmHg). These results indicate that the cuffless BP estimates obtained by the proposed GFSDP algorithm and conventional SVM were in close quantitative agreement with those obtained by the ABP reference values without being overly biased in any direction.

The CIs represent the uncertainties from the GFSDP when estimating cuffless BP. Table 10 shows the lower and upper CI estimations obtained using the GFSDP algorithm. The CIs for the SBP and DBP obtained using the GFSDP with bootstrap algorithm GFSDP$_{BT}$ were smaller than those obtained using conventional methods. The difference between 10.19 mmHg and 11.55 mmHg of the GFSDP and GP algorithm for the SBP and DBP was confirmed. The results of the CI estimation were also shown by applying the Student's t-distribution to the conventional algorithms.

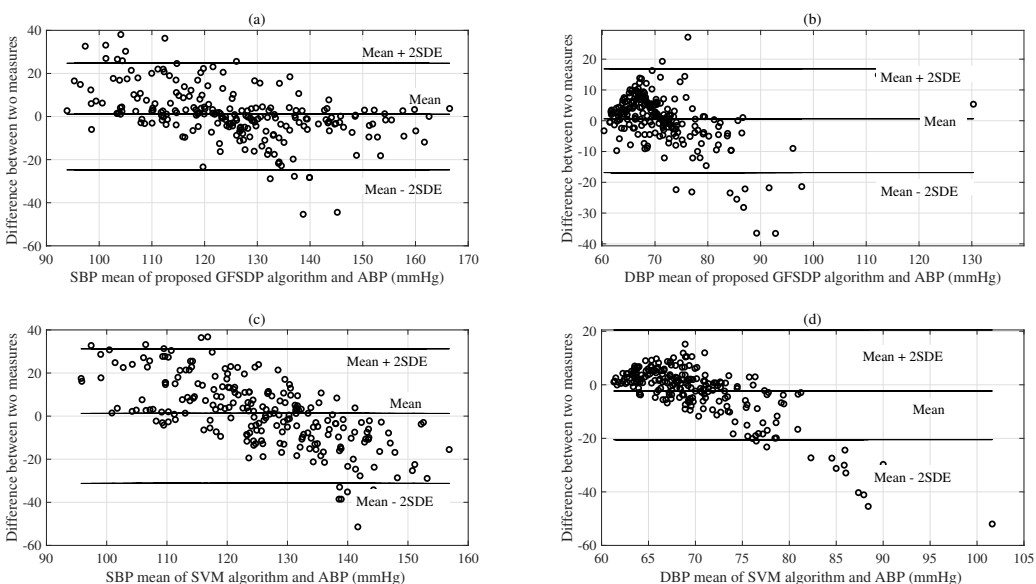

**Figure 5.** We compared the performance between the proposed GFSDP and the reference ABP for the SBP (**a**) and DBP (**b**). We also compared the performance between the SVM and the reference ABP for the SBP (**c**) and DBP (**d**).

## 5. Discussion

This study is the first to propose a GFSDP algorithm to concurrently estimate cuffless BPs and CIs. Table 4 lists the high-scoring features selected using the F-test and RNCA algorithms [38,39]. The ranked features relied on the selection method. They also changed according to the SBP or DBP target variables. Therefore, the final decision process using the AIC algorithm [40] for the proposed hybrid-feature-selected sets was more significant for improving the performance of ML than using a fixed subset.

Table 6 confirms that the proposed GFSDP algorithm was more complex than the GP algorithm in computational complexity. This indicates that computing resources were consumed during the HFD process to weigh the features and finalize the assigned weights. Nevertheless, in terms of estimation accuracy, the proposed GFSDP algorithm exhibited the lowest MAE for the SBP (7.66 mmHg) and DBP (5.47 mmHg) compared with those of the ANN, MLR, SVM, LSTM, and GP algorithms. In particular, compared with the conventional GP algorithm, the accuracy of the SBP and DBP estimation was 30.94% and 22.49%, respectively, confirming that the hybrid feature selection and decision effectively improved the accuracy of estimating SBP and DBP. In addition, the proposed GFSDP algorithm exhibited an improved performance of 48.43% and 32.18% for the SBP and DBP estimations, respectively, compared with the SVM algorithm. The SDEs of the MAEs in all algorithms showed stable values, as shown in Table 7.

The proposed GFSDP algorithm exhibited a slight performance loss compared with the BHS protocol [46], as shown in Table 8. The MAEs were 49.49% ($\leq$5 mmHg), 73.01% ($\leq$10 mmHg), and 84.97% ($\leq$15 mmHg) for the SBP and 60.26% ($\leq$5 mmHg), 85.39% ($\leq$10 mmHg), and 93.74% ($\leq$15 mmHg) for the DBP, as shown in Table 8. Therefore, the proposed GFSDP obtained classes C and B for evaluating SBP and DBP. Furthermore, the proposed GFSDP algorithm was more accurate than conventional algorithms for cuffless BP estimation.

The SDE of the ME was also evaluated according to the AAMI protocol [47]. The proposed GFSDP algorithm for the SBP (10.83 mmHg) and DBP (8.31 mmHg) had a lower SDE of the ME than conventional algorithms; however, all algorithms failed to meet the AAMI criteria, as shown in Table 9. Thus, the AAMI standards are stringent. Figure 5 shows the SDE results of the ME of the proposed GFSDP and SVM algorithms. These results indicate that the results of the SVM were slightly more spread out than those of the proposed algorithm. This further proves the superiority of the proposed algorithm, in addition to its low MAE. Moreover, the proposed GFSDP algorithm is more favorable because it correctly estimated errors based on the weighted feature subset. Therefore, the GFSDP process accurately estimated cuffless BP values.

Cuffless BP-measuring devices typically provide single-point predictions without CIs. Therefore, predicting CIs for cuffless BP measurements improves reliability. Thus, this study predicted CIs using the GP and GFSDP algorithms to express uncertainties in cuffless BP estimation. This study extracted four to eight PPG segment waveforms from each patient's record. A small sample size of each patient affected the accuracy of the bootstrap CI estimation. The results confirmed that the CIs obtained from the proposed GFSDP$_{BT}$ for SBP and DBP were also wide. The CIs derived from the conventional algorithm using the Student's t-distribution (ST) were broader than those obtained from the proposed method for both SBP and DBP, as shown in Table 10. The CI estimates based on the ST distribution are adequate when the sample size from each patient is large (at least 30) [8]. The results confirmed that the proposed GFSDP algorithm was more accurate than the conventional algorithms for cuffless BP and CI estimations. As previously mentioned, GP algorithms predict distributions rather than single-response values. Therefore, CIs were provided along with the predicted responses to represent the uncertainty. However, even with the GP algorithm, few samples for each patient resulted in wide CIs [8].

## 6. Conclusions

The proposed method improved the accuracy and stability of BP and CI estimations using the GFSDP algorithm. This algorithm measures uncertainties, such as CIs, error standard deviation, and bias for cuffless SBP and DBP estimations. Features with high weights for feature selection were selected by applying the F-test and RNCA methods. The AIC algorithm was then used to determine the optimal set of features. In addition, despite the small sample size for each patient, the CIs were presented using the bootstrap method to represent the uncertainty in cuffless BP estimation. The proposed GFSDP is the only one that provides BP and CI assessment concurrently for cuffless BP measurements. Furthermore, the proposed GFSDP algorithm can be designed as a framework for improving healthcare monitoring and optimizing clinical decision support systems. Although public datasets were used in the experiments, the scope of this study is limited by the small sample size for each patient. This is because the CIs of all algorithms representing the uncertainty are extensive. The CIs generated by the GP algorithm are also wide owing to the wide SDE of the ME. Future studies should improve the performance of the GFSDP algorithm by reducing the SDE of the ME. Moreover, the complexity of the GFSDP algorithm should be improved for BP and CI estimations.

**Author Contributions:** Conceptualization, S.L.; methodology, S.L. and G.L.; software, S.L.; validation, S.L., C.-H.S. and G.P.J.; formal analysis, C.-H.S.; investigation, C.-H.S.; resources, A.P.S.; data curation, G.P.J.; writing—original draft preparation, S.L.; writing—review and editing, G.P.J. and

G.L.; visualization, A.P.S.; supervision, C.-H.S.; project administration, G.P.J. and A.P.S.; funding acquisition, C.-H.S. All authors have read and agreed to the published version of the manuscript.

**Funding:** This work was supported by the National Research Foundation of Korea (NRF) grant funded by Korea government (MSIT) (No. 2020R1A2C1010405)

**Data Availability Statement:** Please refer to suggested Data Availability Statements in section "MDPI Research Data Policies" at https://archive.ics.uci.edu/ml/datasets/Cuff-Less+Blood+Pressure+Estimation, accessed on 10 March 2022. Upon a reasonable request, the corresponding author can offer a partial code for the study upon completion of all projects.

**Acknowledgments:** The present research has been conducted by the Research Grant of Kwangwoon University in 2021.

**Conflicts of Interest:** The authors declare no conflict of interest.

## Abbreviations

The following abbreviations are used in this manuscript:

| | |
|---|---|
| BPs | Blood pressures |
| CIs | Confidence intervals |
| CVD | Cardiovascular disease |
| ML | Machine learning |
| ECG | Electrocardiogram |
| PPG | Photoplethysmography |
| PTT | Pulse transit time |
| PWV | Pulse wave velocity |
| MLR | Multiple linear regression |
| GP | Gaussian process |
| SVM | Support vector machine |
| ANN | Artificial neural networks |
| LSTM | Long short-term memory |
| RNCA | Robust neighbor component analysis |
| NCA | Neighbor component analysis |
| AIC | Akaike's information criterion |
| ABP | Arterial blood pressure |
| SBP | Systolic blood pressure |
| DBP | Diastolic blood pressure |
| BHS | British hypertension protocol |
| ME | Mean error |
| SDE | Standard deviation of the error |
| MAE | Mean absolute error |
| AAMI | Association for the Advancement of Medical Instrumentation |

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
