# Peer review of "Cuffless Blood Pressure Estimation with Confidence Intervals using Hybrid Feature Selection and Decision Based on Gaussian Process"

_applsci, doi:10.3390/app13021221_

Round 1

Reviewer 1 Report

Comments to the Authors:

The manuscript entitled "Cuff-less Blood Pressure Estimation with Confidence Intervals using Hybrid Feature Selection and Decision based on Gaussian Process" deals with an interesting subject that fits nicely into the scope of the journal. It introduces a novel machine learning method in one of the important medical issues.

Regrettably, the language does not keep up with the positive impression. There are rather obvious weaknesses with respect to grammar, style, and wording that need to be addressed before the manuscript is considered for publication.

Some examples:

1.     In line 36: it is better to add the expansion of the “PPG” abbreviation, as well as, “ECG” in line 38 (at their first mention).

2.     Please unify the “Cuff-less” in the whole paper, both Cuff-less and Cuffless have been used in this paper both the abstract and text.

3.     Few grammatical errors are found throughout the paper: for example, in line 46: features are essential to the “response variable” or “response variability” or “variable responses”.

4.     In line 62: Therefore, we estimate the CIs by applying the bootstrap method [? ] using the results of the GP algorithm.”?” stand for what?

5.     In line 66: replace the “CI” with confidence intervals (CIs) in “Herein, we obtain confidence intervals (CIs) using…” as well as in lines 69 and 72.

6.     The expanded form of another abbreviation “PTT”: pulse transit time (PTT)” has been written several times, as well as for “Akaike’s information criterion (AIC)” please check the whole paper for these issues like this.

In addition, To make your paper easy to understand, you can redesign and complete figure 1 which summarizes the steps and all the influencing features in BP and its CI.

Author Response

Reviewer#1, Concern # 1:  

Regrettably, the language does not keep up with the positive impression. There are rather obvious weaknesses with respect to grammar, style, and wording that need to be addressed before the manuscript is considered for publication.

Author response:  Authors agreed.

Author action: The authors have completed proofreading from native English speakers.

Reviewer#1, Concern # 2:  

  1. In line 36: it is better to add the expansion of the “PPG”abbreviation, as well as, “ECG” in line 38 (at their first mention).

Author response:  Authors agreed.

Author action: We modified it as

“a new long short-term memory and convolutional neural network using cuffless BPs estimation based on the photoplethysmography (PPG) and electrocardiogram (ECG) signals.”                                                   page 2, lines 38-39.

Reviewer#1, Concern # 3:  

  1. Please unify the “Cuff-less” in the whole paper, both Cuff-less and Cufflesshave been used in this paper both the abstract and text.

Author response:  Authors agreed.

Author action: We fixed it as

“on PPG signal-based cuffless BPs estimations”                   page 2, line 44.

Reviewer#1, Concern # 4:  

  1. Few grammatical errors are found throughout the paper: for example, in line 46: features are essential to the “response variable” or“response variability” or “variable responses”.

Author response:  Authors agreed.

Author action: We unified it as a “response variable”

Reviewer#1, Concern # 5:  

In line 62: Therefore, we estimate the CIs by applying the bootstrap method [? ] using the results of the GP algorithm.”?” stand for what?

Author response:  Authors agreed.

Author action: We modified it as

“Therefore, CIs are estimated by applying the bootstrap method [8] using the results of the GP algorithm [31] to represent the uncertainty in the cuffless BP estimation.”

page 2, lines 63-64

Reviewer#1, Concern # 6:  

In line 66: replace the “CI” with confidence intervals (CIs) in “Herein, we obtain confidence intervals (CIs) using…” as well as in lines 69 and 72.

Author response:  Authors agreed.

Author action: We modified it as

“This study obtained CIs using a bootstrap algorithm [10],[35] to determine the uncertainty (physiological variability) in cuffless BP estimation. Specifying CIs for cuffless measurements is beneficial because CIs measurements are necessary for estimating BP.” 

page 2, lines 68-71

Reviewer#1, Concern # 7:  

The expanded form of another abbreviation “PTT”: pulse transit time (PTT)” has been written several times, as well as for “Akaike’s information criterion (AIC)” please check the whole paper for these issues like this.

Author response:  Authors agreed.

Author action: We checked it and made a table for abbreviation as

page 19

Reviewer#1, Concern # 8:  

In addition, To make your paper easy to understand, you can redesign and complete figure 1 which summarizes the steps and all the influencing features in BP and its CI.

Author response:  Authors agreed.

Author action: we redesigned Fig. 1 as   

page 3

Reviewer 2 Report

1.     The authors have applied traditional machine learning modes for Cuff-less Blood Pressure Estimation. However, previous studies utilizing advanced deep learning models with confidence intervals and feature selection/decision also exists for the Cuff-less Blood Pressure Estimation. Therefore, the reviewer fails to perceive enough novelty of the research for publishing in the prestigious Applied Sciences journal. Consequently, the reviewer suggests its rejection.

2.     Moreover, the following points/lines also require correction/revision:

i)      On line 157 (page 5) and line 310 (page 15), LSTM has been mentioned but not used in Tables, Introduction or Discussion sections, etc. However, the comparison of results based on LSTM is also required to contrast deep learning models with machine learning models.

ii)    Some of the references are very old and may be revised with new research (where possible)

iii)  Line 62 (page 2), line 266 (page 14), unexplained question mark (?)

iv)   Word correction is required (Line 182 of page 8)

v)     Extra spacing line 42 (page 2), line 283 (page 14), line 312 (page 15)

Author Response

Reviewer#2, Concern # 1:  

  1. The authors have applied traditional machine learning modes for Cuff-less Blood Pressure Estimation. However, previous studies utilizing advanced deep learning models with confidence intervals and feature selection/decision also exists for the Cuff-less Blood Pressure Estimation. Therefore, the reviewer fails to perceive enough novelty of the research for publishing in the prestigious Applied Sciences journal. Consequently, the reviewer suggests its rejection.

Author response:  The authors appreciate the review of this paper. However, we disagree with the above review.

The first reason is that deep learning can only be superior to conventional machine learning for some data. In other words, the authors believe that different results can occur depending on the data.

The first author has conducted blood pressure estimation and confidence interval studies since 2009.

However, as far as I know, the first author published a paper around 2011 on a confidence interval study in oscillometric-based blood pressure estimation, and uncertainty estimation studies were published around the same time as follows:

[3] S. Lee, M. Bolic, V. Groza, H. Dajani, and S. Rajan, Confidence interval estimation for oscillometric blood pressure measurements using bootstrap approaches, IEEE Transactions on Instrumentation and Measurement, vol. 60, no. 10, pp. 3405-3415, Oct. 2011 (IF:3.658, Top 14.1%)

[8] S. Lee, Improved confidence interval estimation for oscillometric blood pressure measurement by combining bootstrap-after-jackknife function with non-Gaussian models, Mathematical Problems in Engineering, vol. 2014, article ID 231925, pp. 1-10, Nov. 27. 2014 (IF:1.009)

[12] S. Lee, S. Rajan, C.-H. Park, J-.H. Chang, H. Dajani, and V. Groza, Estimated confidence interval from single pressure measurement based on algorithmic fusion, Computers in Biology and Medicine, vol. 62, pp. 154-163, Jul. 2015 (IF:3.434, Top13.6 %)

[17] S. Lee, and J.-H. Chang, Oscillometric Blood Pressure Estimation Based on Deep Learning, IEEE Transactions on Industrial Informatics, vol. 13, no. 2, pp. 461-472, Apr. 2017 (IF:9.112, Top 4.16%) 

[26] S. Lee, and G. Lee, Ensemble methodology for confidence interval in oscillometric blood pressure measurements, Journal of Medical Systems44, 91, pp. 1-9, Mar. 2020 (IF: 3.058, Top 22.5%)

[28] S. Lee, H. Dajani, S. Rajan, G. Lee, V.  Groza, Uncertainty in Blood pressure measurement estimated using ensemble-based recursive methodology, SENSORS, vol. 20 (7), 2108, Apr. 2020 (IF:3.275, Top 23.4%)  

Dieterle, T.;   Battegay, E.;    Bucheli, B.;    Martina,  B. Accuracy and `range of uncertainty’ of oscillometric blood pressure monitors around the upper arm and the wrist. Blood Press Monit. 1998, 3, 339–346.

Soueidan, K.; Chen, S.; Dajani, H. R.;  Bolic, M.;  Groza, V. Augmented blood pressure measurement through the noninvasive estimation of physiological arterial pressure variability, Physiological measurement. 2012, 33, 881-899.

However, we have yet to find a study to estimate the confidence interval in the recently popular PPG- based blood pressure estimation study.

Author action:

Accordingly, this study proposed a new methodology for simultaneously estimating blood pressure (BP) and confidence intervals (CIs) using cuffless BP measurement through hybrid feature selection and decision based on the Gaussian process. This study aimed to estimate BP and CIs, representing the uncertainty of cuffless BP estimation.

Reviewer#2, Concern # 2:  

  1. Moreover, the following points/lines also require correction/revision:
  2. i)On line 157 (page 5) and line 310 (page 15), LSTM has been mentioned but not used in Tables, Introduction or Discussion sections, etc. However, the comparison of results based on LSTM is also required to contrast deep learning models with machine learning models.

Author response:  Authors fully agreed.

Author action: The authors experimented with LSTM and compared the results in Tables 5-10.  

“This study used the ANN, MLR, SVM, LSTM, and GP algorithms [17], [31] as conventional algorithms to evaluate the performance of the proposed GFSDP algorithm.

~~

The LSTM algorithm shows the MAEs of the SBP (10.78 mmHg) and DBP (6.98 mmHg) compared with the reference BP.”        

page 15, line 291

Reviewer#2, Concern # 3:  

  1. ii)Some of the references are very old and may be revised with new research (where possible)

Author response:  Authors agreed.

Author action: The author recently included related papers as follows:

  1. Lee, S.; Dajani, H.; Rajan, S.; Groza, V. Uncertainty in Blood pressure measurement estimated using ensemble-based recursive methodology. Sensors.  2020,  20 (7),  1-15.
  2. Lee, S.; Lee, G. Ensemble methodology for confidence interval in oscillometric blood pressure measurements. Journal of Medical Systems.  2020,  44 (91),  1-9.
  3. Noor, F. A.; Mohamed, A. LSTM Multi-Stage Transfer Learning for Blood Pressure Estimation Using Photoplethysmography. Electronics. 2022,  11, 1-17.
  4. Valeria, F.; Sofia, G.; Daniele, R.; Ilenia, C.; Monica, V.;  Guido, P.  Improving Cuff-Less Continuous Blood Pressure Estimation with Linear Regression Analysis.  Electronics.  2022,  11 (19), 1-18.
  5. Simjanoska, M.; Gjoreski, M.; Gams, M.; Bogdanova, A.M. Non-Invasive Blood Pressure Estimation from ECG Using Machine Learning Techniques. Sensors. 2018,  18, 1160.
  6. Maqsood, S.; Xu, S.; Springer, M.; Mohawesh, R. A Benchmark Study of Machine Learning for Analysis of Signal Feature Extraction Techniques for Blood Pressure Estimation Using Photoplethysmography.  IEEE Access.  2021,  9,  138817 - 138833.
  7. Jared Leitner, J.; Chiang, P.-H.; Dey, S. Personalized Blood Pressure Estimation Using Photoplethysmography: A Transfer Learning Approach. IEEE Journal of Biomedical and Health Informatics.  2022,   26 (1),  218 - 228.
  8. Vazquez, E. A.; Ewert, D.; Jorgenson, D.;   Sand, M. Assessment of the Uncertainty Associated With Two Consecutive Blood Pressure Measurements Using the Auscultatory Method. IEEE Transactions on Instrumentation and Measurement.   2022,  71, 1-11.

Reviewer#2, Concern # 4:  

iii)  Line 62 (page 2), line 266 (page 14), unexplained question mark (?)

Author response:  Authors agreed.

Author action: We fixed it as

“Therefore, CIs are estimated by applying the bootstrap method [8] using the results of the GP algorithm [31] to represent the uncertainty in the cuffless BP estimation.”

page 2, lines 63-64

“The probability of the British Hypertension Society (BHS) protocol [45] was also obtained based on the results of the MAE and SDE, as shown in Table 8.”

page 14, line 269

Reviewer#2, Concern # 5:  

  1. iv)Word correction is required (Line 182 of page 8)

Author response:  Authors agreed.

Author action: We corrected it as

“Feature selection involves selecting essential features from the original feature set. This implies that only a few features affect the target BPs. Thus, reducing the dimensions of the feature space while retaining only valid information is essential for BP estimation.”

page 14, line 269

Reviewer#2, Concern # 6:  

  1. v)Extra spacing line 42 (page 2), line 283 (page 14), line 312 (page 15)

Author response:  Authors agreed.

Author action: We checked it and reviewed and reformatted every sentence in each section.

Reviewer 3 Report

Attached

Author Response

Reviewer#3, Concern # 1:  

  • The first sentence requires a reference. “It was reported that by 2019, there were 523 million people with cardiovascular disease and 18.6 million deaths.”

Author response:  Authors fully agreed.

Author action: Reference papers have been added as follows:

“By 2019, 523 million people have had a cardiovascular disease, and 18.6 million related deaths have been reported [1].”

page 1, line 18

Reviewer#3, Concern # 2:  

There are some transition sentences that are inappropriate for logical flow of the study.

o In line 25, authors mention about the physiological parameters’ importance, and in Line 27, they are skipping protocols for confidence intervals. It is hard to understand this transition sentence.

Author response:  Authors agreed.

Author action: We modified the sentence as

“Rapid changes in BP in these patients can indicate an underlying severe illness. Furthermore, BP varies owing to intrinsic physiological changes for various reasons, such as food intake, environmental temperature, exercise, disease, and stress. Thus, the precision and uncertainty of BP measurements induced by physiological parameters [4] have been a constant concern for clinicians and practitioners [5], [6], [7]. However, using BP monitoring devices for estimating BP uncertainty is currently impossible. This necessitates a standard protocol for confidence intervals (CIs), representing the uncertainty of BP monitors [5], [8], [9].”

page 1, lines 22~

Reviewer#3, Concern # 2:  

o What is the previous critical issue for increasing the reliability of ML algorithms? Why do authors start with “another”. “Another critical issue for increasing the reliability of ML algorithms is determining which features are essential to the response variable.”

Author response:  Authors agreed.

Author action: We changed the sentences and rewrote some sentences as follows.

“~Many studies on PPG signal-based cuffless BP estimations have been conducted [13], [14], [21],[22], [23]. By contrast, studies on CIs estimation are limited. However, there have been studies on the uncertainty of estimating a few BP data based on conventional oscilloscope BP measurements [5], [6],[7],[8],[9]. The two most used methods for cuffless BP estimation were obtained using the extracted features and pulse transit time (PTT) from PPG signal pulses [15], [24],[25],[26].”~

page 2, lines 44~

Reviewer#3, Concern # 3:  

  • Authors says “The ML algorithms, including multiple linear regression (MLR), artificial neural network (ANN), and support vector machine (SVM) [13], were utilized to estimate BP [11].” But they only give one reference for SVM, not other ML algorithms.

Author response:  Authors agreed.

Author action: We modified the sentences and rewrote some sentences as

“Machine learning (ML) algorithms are commonly used for BP estimation [5],[9],[10],[11],[12]. ML algorithms, including multiple linear regression (MLR) [14], [15], artificial neural networks (ANN) [13], [15], [16], and support vector machine (SVM) [17],[18],[19] have been utilized for estimating BP [15], [20], [21], [22].”

page 1, lines 33~

Reviewer#3, Concern # 4:  

  • Authors have only given a refence about PPG signals and concluded that there are many studies on PPG signal and none of them estimated CIs. How can readers convince that? “Although there have been many studies on PPG signal-based cuff-less BP estimation, no previous research has estimated CIs, representing uncertainty for BP.”

Author response:  Authors agreed.

Author action: We fixed the sentences and rewrote some sentences as

“~Many studies on PPG signal-based cuffless BP estimations have been conducted [13], [14], [21],[22], [23]. By contrast, studies on CIs estimation are limited. However, there have been studies on the uncertainty of estimating a few BP data based on conventional oscilloscope BP measurements [5], [6],[7],[8],[9]. The two most used methods for cuffless BP estimation were obtained using the extracted features and pulse transit time (PTT) from PPG signal pulses [15], [24],[25],[26].”~

page 2, lines 44~

Reviewer#3, Concern # 5:  

  • I recommend putting citations after author name like “P. Nandi et al. [18] proposed” in Line 36, not in Line 35, 40, ….

Author response:  Authors agreed.

Author action: We changed it as  

“Wang et al. [13] introduced a BP estimation method using photoplethysmography (PPG) signals using novel ANNs.

Nandi et al. [23] proposed a new long short-term memory (LSTM) and convolutional neural network using cuffless BPs estimation based on PPG and electrocardiogram (ECG) signals.

Multichannel PPG was introduced using an SVM ensemble-based continuous BPs estimation in [19]. Qiu et al. [15] proposed a new method for estimating BP using a window function-based piecewise neural network.”

page 2, lines 36~41

Reviewer#3, Concern # 6:  

  • Between Line 90 and Line 104, authors clearly indicate study’s aim, outcomes, and contribution. Because of lack of information about the related work, it is not clear to me to support contribution and originality of the study.

Author response:  Authors fully agreed.

Author action: We included the related research and represented the study’s aim and outcomes, and contribution as

“Accordingly, this study proposed a new methodology for simultaneously estimating BP and CIs using cuffless BP measurement through hybrid feature selection and decision based on the Gaussian process (GP). This study aimed to estimate BP and CIs, representing the uncertainty of cuffless BP estimation.”

page 2, lines 54~56

“The proposed hybrid feature selection and decision based on the GP algorithm can provide a means for distinguishing between estimation errors (statistical variance of estimates) and changes in estimates owing to physiological variability [5], [7].”

page 2, lines 66~68

“This study provides uncertainty for cuffless BP measurements and introduces a method for reducing the error of BP estimates based on PPG and ECG signals. As previously mentioned, the proposed study estimates the exact BPs and CIs concurrently, representing the uncertainty of cuffless BP estimation.  To the best of the authors’ knowledge, this study is the first to propose a GP-based feature selection and decision process (GFSDP) algorithm to simultaneously estimate cuffless BPs and CIs. Although some CI studies on conventional oscillometric BP estimation methods have been performed [5],[6],[7],[8], studies on CI estimation in cuffless BP measurements are limited, as summarized in Table 1. The contributions of the study to BP and CI estimation are as follows:

  • CIs are estimated using a bootstrap based on the GP algorithm to express uncertainty in cuffless BPs estimation.
  • The proposed methodology uses a hybrid F-test and RNCA to select the weighted features among the original features.
  • An adaptive AIC of automatically calibrated likelihoods is proposed based on the GP algorithm to determine the best feature subset as a model selection problem.”

page 2, lines 95~109

Reviewer#3, Concern # 7:  

I recommend creating a literature table summarizing the previous studies and explaining them to show your motivation as a result of this table. It will also support your best knowledge in “To the best of the authors’ knowledge, this is the first study of an HFDGP algorithm” in Line 96.

Author response:  Authors fully agreed.

Author action: We included literature summarizing as in Table 1.

Reviewer#3, Concern # 8:  

  • Give a space before citations. “variability[6].” In Line 66

Author response:  Authors agreed.

Author action: We fixed it and changed the sentence as  

“The proposed hybrid feature selection and decision based on the GP algorithm can provide a means for distinguishing between estimation errors (statistical variance of estimates) and changes in estimates owing to physiological variability [5],[7].”

page 2, lines 66~68

Reviewer#3, Concern # 8:  

  • Which studies did try? “However, only some researchers have tried to determine the uncertainty of physiological measurements.”

Author response:  Authors agreed.

Author action: We changed the sentence as  

“However, studies on determining the uncertainty of physiological measurements [4] using oscillometric BP signals [5],[6],[7],[8] are limited.”

page 2, lines 74~76

Reviewer#3, Concern # 8:  

  • Please prefer a scientific language instead of subject or possessive pronouns (we, our, …)

Author response:  Authors agreed.

Author action: We have rewritten that sentence in the introduction as

“The proposed hybrid feature selection and decision based on the GP algorithm can provide a means for distinguishing between estimation errors (statistical variance of estimates) and changes in estimates owing to physiological variability [5],[7].”

page 2, lines 66~68

“This study provides uncertainty for cuffless BP measurements and introduces a method for reducing the error of BP estimates based on PPG and ECG signals. As previously mentioned, the proposed study estimates the exact BPs and CIs concurrently, representing the uncertainty of cuffless BP estimation.  To the best of the authors’ knowledge, this study is the first to propose a GP-based feature selection and decision process (GFSDP) algorithm to simultaneously estimate cuffless BPs and CIs. Although some CI studies on conventional oscillometric BP estimation methods have been performed [5],[6],[7],[8], studies on CI estimation in cuffless BP measurements are limited, as summarized in Table 1. The contributions of the study to BP and CI estimation are as follows:

  • CIs are estimated using a bootstrap based on the GP algorithm to express uncertainty in cuffless BPs estimation.
  • The proposed methodology uses a hybrid F-test and RNCA to select the weighted features among the original features.
  • An adaptive AIC of automatically calibrated likelihoods is proposed based on the GP algorithm to determine the best feature subset as a model selection problem.”

page 2, lines 95~109

Reviewer#3, Concern # 9:  

Figure 1 represents the flow of the proposed methodology. I recommend making connections in the case study authors describe. At first glance, it is hard to understand what was done for which step in Figure 1.

Author response:  Authors agreed.

Author action: We included a detail describing the step in Fig. 1 and redesigned Fig. 1 as

“Fig. 1 shows the block diagram of the GFSDP algorithm. The first stage was obtained using PPG and ECG signals, as shown in Fig. 1 (a). At this stage, a public dataset was acquired from the University of California Irvine (UCI) ML repository center [28]. Outliers were removed in the preprocessing stage. Valuable features extracted after preprocessing are essential for accurate BP and CI estimations from ECG and PPG signals, as shown in Fig. 1 (c). Subsequently, the hybrid F-test [39] and RNCA [40] algorithms were used to select weighted features from the original features, as shown in Fig. 1 (d). Thereafter, the adaptive AIC determined the best feature subset as a model selection problem using likelihoods based on the GP algorithm, as shown in Fig. 1 (e). Finally, the best feature subset was used as the input feature for the GP algorithm.”

page 3, lines 111~120

Reviewer#3, Concern # 10:  

  • Authors uses so many abbreviations. Some of them are used only once. In my opinion, there is no need to define abbreviations for the phrases that were not used again during the text. For the remaining abbreviations, creating an abbreviation table may help reader. Abbreviations have to be defined in the first occurrence. For example, authors use AIC in Line 161 but they give the long form of AIC in Line 166. On the other hand, authors describe RNCA in Line 84, but they give its long form and abbreviation again in Line 165. All abbreviations should be checked.

Author response:  Authors agreed.

Author action: We made an abbreviation table and checked all abbreviations as

page 19

Round 2

Reviewer 2 Report

Although similar studies exist utilizing nearly the same methodology as proposed in this paper. However, the authors have substantially improved the revised version of the manuscript.
Therefore, the reviewer recommends its publication.